# Enhanced Carotid Plaque Echolucency Is Associated with Reduced Cognitive Performance in Elderly Patients with Atherosclerotic Disease Independently on Metabolic Profile

**DOI:** 10.3390/metabo13040478

**Published:** 2023-03-27

**Authors:** Daniela Mastroiacovo, Alessandro Mengozzi, Francesco Dentali, Fulvio Pomero, Agostino Virdis, Antonio Camerota, Mario Muselli, Stefano Necozione, Raffaella Bocale, Claudio Ferri, Giovambattista Desideri

**Affiliations:** 1Angiology Unit, Medical Department, “SS. Filippo and Nicola” Hospital, Avezzano, 67051 L’Aquila, Italy; 2Department of Clinical and Experimental Medicine, University of Pisa, 56126 Pisa, Italy; 3Center for Translational and Experimental Cardiology (CTEC), Department of Cardiology, Zurich University Hospital, University of Zurich, 8952 Schlieren, Switzerland; 4Institute of Life Sciences, Scuola Superiore Sant’Anna, 56126 Pisa, Italy; 5Department of Medicine and Surgery, Insubria University, 21100 Varese, Italy; 6Department of Internal Medicine, Michele and Pietro Ferrero Hospital, Verduno, 12060 Cuneo, Italy; 7Department of Life, Health and Environmental Sciences, University of L’Aquila, 67100 L’Aquila, Italy; 8Division of Endocrine Surgery, Agostino Gemelli University Hospital Foundation IRCCS, Catholic University of the Sacred Heart, 00168 Rome, Italy

**Keywords:** cognitive functions, carotid atherosclerotic plaque, grayscale median

## Abstract

Vulnerable carotid atherosclerotic plaques are related to an increased risk of cognitive impairment and dementia in advanced age. In this study, we investigated the relationship between the echogenicity of carotid plaques and cognitive performance in patients with asymptomatic carotid atherosclerotic plaques. We enrolled 113 patients aged 65 years or more (72.4 ± 5.9 years) who underwent carotid duplex ultrasound to evaluate plaque echogenicity by grey-scale median (GSM) and neuropsychological tests to assess cognitive function. The GSM values at baseline were inversely correlated with the number of seconds required to complete Trail Makin Test (TMT) A (rho: −0.442; *p* < 0.0001), TMT B (rho: −0.460; *p* < 0.0001) and TMT B-A (rho: −0.333; *p* < 0.0001) and directly correlated with Mini Mental State Examination (MMSE) and Verbal Fluency Test (VFT) score (rho: 0.217; *p* = 0.021 and rho: 0.375; *p* < 0.0001, respectively) and the composite cognitive z-score (rho: 0.464; *p* < 0.0001). After a mean period of 3.5 ± 0.5 years, 55 patients were reevaluated according to the same baseline study protocol. Patients with baseline GSM value higher than the median value of 29 did not show any significant variation in the z-score. Instead, those with GSM ≤ 29 showed a significant worsening of z-score (−1.2; *p* = 0.0258). In conclusion, this study demonstrates the existence of an inverse relationship between the echolucency of carotid plaques and cognitive function in elderly patients with atherosclerotic carotid disease. These data suggest that the assessment of plaque echogenicity if used appropriately, might aid in identifying subjects at increased risk for cognitive dysfunction.

## 1. Introduction

Extracranial asymptomatic carotid artery stenosis is a common finding among the elderly population and represents a well-established risk factor for ischemic stroke, contributing to 10–20% of strokes or transient ischemic attacks [1]. Indications for surgical intervention (carotid endarterectomy or stenting) in hemodynamically relevant carotid artery stenosis are primarily based on the degree of stenosis. Revascularization is strongly recommended in symptomatic internal carotid plaques in order to remove a potential source of the embolism and to reduce the risk of recurrent stroke and death [2]. Conversely, the optimal management of asymptomatic carotid stenosis is still the object of debate [3]. Indeed, although the improvements in best medical treatment have contributed to reducing the number of cerebrovascular events among asymptomatic patients [4], in the last decades, several studies have highlighted the need to improve risk stratification and optimize therapy in selected high-risk patient subgroups [5,6]. Current guidelines suggest extending the assessment of asymptomatic carotid stenosis beyond the size of plaque and grade of stenosis [2]. In fact, morphological plaque characterization and evaluation of parameters such as calcification, vascularization, echolucency and surface may also have some prognostic and therapeutic implications, although not all of them are sufficiently validated [7,8]. In particular, echolucent plaques are actually supposed to be more at risk of embolization than iso- or hyperechogenic plaques because they usually correspond to lipid deposits, intraplaque necrosis, or hemorrhage [9,10].

An increasing controversy relating to asymptomatic carotid stenosis is its potential involvement in the pathophysiology of cognitive impairment and dementia. There are several potential mechanisms whereby asymptomatic carotid stenosis may cause cognitive impairment directly, including silent cerebral infarction, silent embolization, involvement in the pathophysiology of white matter hyperintensities or a hemodynamic etiology [11]. A recent meta-analysis reported a statistically significant association between asymptomatic carotid plaques and one or more tests suggesting cognitive impairment in 94% of the studies included, although the authors found significant heterogeneity, which compromised the overall interpretation of the results [12].

The gray-scale median (GSM) analysis is a standardized and reproducible method that, considering the median value of the whole atherosclerotic plaque, allows to perform a quantitative and objective assessment of the echogenicity of the plaque by B-mode ultrasound and digital image examination [13,14,15,16].

In this study, using the GSM analysis, we investigated the association between the echogenicity of carotid plaques and cognitive performance in patients with carotid atherosclerotic plaques and no history of cerebrovascular events and/or clinical evidence for dementia.

## 2. Materials and Methods

### 2.1. Subjects

Participants were recruited among those with a history of known or suspected carotid artery atherosclerotic disease who were consecutively referred to our Angiology Unit after ultrasonographic demonstration of atherosclerotic plaques defined as at least a focal thickness > 1.5 mm as measured from intima-lumen to media-adventitia interfaces [17]. Exclusion criteria included age < 65 years, history and symptoms or signs of neurological disease and cerebrovascular events (transient ischemic attack or stroke) or neuroradiological evidence, if available, of vascular brain lesions, previous carotid endarterectomy, previous neck irradiation, presence of major depressive states, psychiatric disorders and overt cognitive dysfunction. All participants underwent a complete clinical evaluation, including a comprehensive medical history, particularly for the assessment of the individuals’ vascular risk factors, a general and neurological examination, blood sampling and neuropsychological tests. Before neuropsychological testing, clinical systolic and diastolic blood pressures were recorded in the morning with the use of a validated oscillometric device with appropriately sized cuffs (Omron HEM 7155-E; Omron Matsusaka Co. Ltd., Kyoto, Japan) on the non-dominant upper arm. Within 24 h of neuropsychological testing, blood samples were drawn from each participant after an overnight fasting period for routine serum biochemistry tests, including lipid profile and fasting plasma glucose. All participants underwent ultrasound evaluation of neck vessels to assess the degree of stenosis and plaque echogenicity. All patients with carotid plaques causing severe stenosis (70% or more according to the NASCET method) were candidates for endarterectomy and underwent CT brain scans before the surgical procedure in order to exclude recent ischemic lesions. Starting from the third year after the first evaluation, participants were re-evaluated according to the same procedures used at baseline.

### 2.2. Cognitive Function Assessment

Neuropsychologically trained research assistants, who were unaware of the results of ultrasound examinations of the neck vessels, tested all the participants during a morning visit in a quiet room. Cognitive testing was performed using combinations of several standardized tests that included the Mini Mental State Examination (MMSE), Trail Making Test (TMT) A, TMT B and Verbal Fluency Test (VFT). The MMSE was used to assess global cognitive function and contained 18 items that explore orientation, memory, attention, ability to follow commands and copying a complex image, with a score range from 0 to 30. The mean MMSE score for cognitively intact patients is 27 points; a score of 24 points or lower indicates some degree of cognitive impairment [18]. TMT is a frequently used neuropsychological test because of its sensitivity to brain damage [19]. It explores visual-conceptual and visual-motor tracking. TMT is administered in two parts. Part A is a visual-scanning, timed task where participants are asked to connect with lines 25 circles numbered from 1 to 25 as quickly as possible. The test is terminated after 5 min even if not completed. In Part B, participants are asked to connect circles containing numbers (from 1 to 13) or letters (from A to L) in an alternate numeric/alphabetical order. The test is terminated in every case after 10 min even if not completed. The TMT B minus TMT A score, calculated as the difference between TMT B and TMT A times, is considered a measure of cognitive flexibility, relatively independent of manual dexterity. VFT is a short test of verbal functioning [20]. Participants were given 60 s to produce as many unique words as possible, starting with a given letter. The participant’s score is the number of unique correct words. An integrated measure of overall cognitive function—composite cognitive z-score—was also constructed for each participant by converting the log-transformed raw scores from the individual tests to standardized scores (z-score) that were based on the means and pooled SDs of the whole cohort at baseline.

### 2.3. Ultrasound Method

The ultrasound examinations were performed by an experienced sonographer using a Siemens Acuson Sequoia 512 ultrasound machine (Siemens Healthcare s.r.l., Milano, Italy) and a 7–10 MHz high-frequency linear probe. All patients were examined supine with a slight head tilt. Anterior, lateral and posterior projections were used to image the plaque longitudinally. The largest plaque visualized with an optimal projection at the carotid bifurcation or at the proximal internal carotid artery was chosen for the assessment of plaque echogenicity. In the case of hypoechoic or anechoic plaques, one image with color Doppler or power Doppler was saved to ensure the correct delineation of the plaque margin. The settings for the ultrasound machine were adjusted and standardized for all examinations by using a maximum dynamic range (60 dB) and by setting the gain to ensure an almost noiseless vessel lumen (blood) and an echo-attenuated area of adventitia. B-mode and corresponding color-doppler images were saved in digital form on a magneto-optical disk. All appropriate images were stored in and processed by a computer. Additionally, the degree of stenosis based on B-mode images was measured according to the NASCET method [7].

### 2.4. Plaque Echogenicity

The GSM analysis was performed by the same operator, who was blinded to the clinical profile of the patients. Images were analyzed using a graphics program (Adobe Photoshop 5.0, Adobe Systems Incorporated) with a 2- to 3-fold increase in initial size. According to a previously described and validated methodology [21,22], the color information in the digitized image was omitted so that all processing and analysis were performed on images in gray mode. The linear scale of the “curves” option of the software was adjusted to achieve values of the blood between 0 and 5 and gray values of the adventitia between 185 and 195. In these normalized images, each plaque was outlined using the computer mouse, and its grayscale content was analyzed for the mean, standard deviation, median and total pixel calculation using the histogram facility. The GSM, which represents the median of the frequency distribution of tones of pixels included in the plaque areas, was used as a measure of the overall plaque echogenicity.

### 2.5. Statistical Analysis

Wilcoxon rank-sum test was used for unpaired continuous variables comparison. Paired comparisons between continuous variables were performed using Wilcoxon signed-rank test. Spearman’s rank correlation was used to assess relationships between variables. Multivariate linear regression analysis was conducted to identify independent factors correlated with global cognitive functioning (z-score). A two-factor analysis of variance (ANOVA) with repeated measures on one factor (time) was performed to evaluate differences in z-score over time between two groups created on GSM median value. A *p*-value < 0.05 was considered statistically significant when comparing variables. All the statistical analysis was performed using STATA 17.0 software (College Station, TX, USA: StataCorp LLC). The estimated sample was calculated using the Medcalc^®^ Statistical Software version 20.126 (MedCalc Software Ltd., Ostend, Belgium). Setting the type I error to 5%, the statistical power to 80% and the expected correlation coefficient to 0.3, the minimum number of participants who should have been included was 112.

## 3. Results

One hundred and thirteen consecutive elderly patients (72.4 ± 5.9 years, 54 females, 59 males) were recruited. General characteristics of the study population are shown in Table 1. Hypertension and hypercholesterolemia were the most prevalent cardiovascular risk factors (80.5% and 73.5%, respectively).

The GSM values were inversely correlated with the number of seconds required to complete TMT A (rho: −0.442; *p* < 0.0001), TMT B (rho: −0.460; *p* < 0.0001) and TMT B-A (rho: −0.333; *p* < 0.0001) and directly correlated with MMSE (rho: 0.217; *p* = 0.021) and VFT (rho: 0.375; *p* < 0.0001) scores and the composite cognitive z-score (rho = 0.464; *p* < 0.0001) (Figure 1). An inverse correlation between GSM and NASCET score was also found (rho: −0.214; *p* = 0.023). The linear regression model showed a significant association of GSM values with age and z-score independent of the other variables considered (Table 2). Neither serum lipid parameters nor blood pressure levels were correlated with neuropsychological test scores and GSM values (Table A1). None of these parameters influenced the relationship between GSM values and cognitive performance (Table A2). Significant correlations between NASCET values and both systolic and diastolic blood pressure were found at baseline (Table A1), but the independence of these associations was not confirmed by multiple linear regression analysis (systolic blood pressure β = 0.221, *p* = 0.137; diastolic blood pressure β = 0.264, *p* = 0.358). When the study population was divided into two groups according to the median GSM value (≤29 and >29, respectively), all the neuropsychological tests scores, other than MMSE, and the composite z-score were found to be significantly worse in a patient with low GSM score (Table 3).

During the follow-up, 32 patients undergone carotid endarterectomy and were excluded from the follow-up. Among the 81 patients kept on medical treatment, six died (five cardiovascular deaths), eight experienced cerebrovascular or cardiovascular events, and 12 declined to attend the follow-up evaluation. Thus, the longitudinal analysis was performed in the remaining 55 patients (33 females and 22 males) after a mean follow-up period of 3.5 ± 0.5 years. GSM value did not significantly change during follow-up (32 ± 11.8 vs. 31.9 ± 11.1, *p* = 0.752) as also MMSE, TMT A and VFT scores and the composite cognitive z-score while TMT B and TMT B-A scores significantly increased (Table 4). A significant variation of NASCET values was also observed in the subgroup of patients followed during the longitudinal phase of the study (from 32.1 ± 12.1 to 36.9 ± 13.1, *p* < 0.0001).

The GSM values were still inversely correlated with the number of seconds required to complete TMT A (rho: −0.347; *p* = 0.009), TMT B (rho: −0.395; *p* = 0.003) and TMT B-A (rho: −0.381; *p* = 0.004) and directly correlated with the scores of MMSE (rho: 0.314; *p* = 0.020), VFT (rho: 0.319; *p* = 0.018) and composite cognitive z-score (rho: 0.392; *p* = 0.003) (Figure 2A). On the contrary, the significant correlation between GSM and NASCET we found at baseline was no longer evident at follow-up (rho: −0.254; *p* = 0.062). The linear regression model showed that also in this latter analysis, only the GSM value and age were significantly associated with a better z-score, independently of the other variables considered (Table 5). The correlations between NASCET values and both systolic and diastolic blood pressure observed at baseline were lost at follow-up.

At the follow-up examination, we evaluated differences in cognitive performance measured with z-score separately into two groups based on a median GSM value of 29 assessed at the first examination. Those with GSM > 29 (n = 31) did not show any significant variation in the z-score. Instead, those with GSM ≤ 29 (n = 24) showed a significant worsening of z-score (−1.2; *p* = 0.0258) (Figure 2B). Indeed, the repeated-measures ANOVA for z-score showed a significant difference between the 2 GSM groups (*p* = 0.026): in time-to-group interaction, those who had GSM ≤ 29 worsened significantly compared to the other group. The difference between z-score values observed at baseline between GSM groups in the whole study population was no more significant in this sub-cohort of patients while it was newly evident at the end of follow-up.

## 4. Discussion

The current report demonstrates a significant inverse relationship between predominantly echolucent (compared with predominantly echogenic) plaques and cognitive performances in elderly patients with carotid atherosclerosis. To the best of our knowledge, this study is the first one to demonstrate such an intriguing association.

The question of whether some degree of cognitive impairment is an inevitable part of aging or should be considered a pathological pre-stage of dementia is debatable. Aging is usually accompanied by deficits in several domains of cognition, including speed of processing, working memory capacity, inhibitory processes and long-term memory, whereas other aspects of cognitive function, such as implicit memory and knowledge storage, are less influenced by aging [23]. With an aging world population that is ever-increasing, there is a growing need to identify factors influencing cognitive dysfunction and effective preventive and treatment strategies.

One of the factors that seem to play a pivotal role in the pathophysiology of cognitive impairment is an atherosclerotic carotid disease, including flow-limiting stenosis, increased intima-media thickness and high-risk carotid plaque features [24,25]. There has been increased attention to individual plaque components as well as the degree of stenosis as potential determinants of atherosclerotic cardiovascular disease [26,27]. Though there is a strong association between high-risk plaque elements and future and recurrent stroke [28], there are fewer studies evaluating the association of plaque features with the development of cognitive dysfunction and dementia. For example, Auperin et al. found that men aged 59 to 71 years with more plaques were more likely to have poor performance on some neuropsychological tests [29]. Other studies have shown no significant difference in cognitive function when accounting for other cardiovascular risk factors [30].

The direct association between carotid plaque echolucency and poor cognitive performance demonstrated in elderly subjects with asymptomatic carotid atherosclerosis and no evidence of overt cognitive dysfunction at baseline sheds new light on this interesting topic. Notably, this association was unaffected by traditional cardiovascular risk factors known to be shared between cognitive dysfunction and atherosclerotic cardiovascular disease [31]. This finding might allow speculation about a possible direct role of carotid echolucent plaque in the pathophysiology of cognitive dysfunction.

In this regard, the GSM of B-mode ultrasound images is a computer-assisted grading of the echogenicity of atherosclerotic plaques. It is a measure of overall plaque echogenicity, which is a quantitative index of the echoes from the plaque. The GSM has been suggested as a possible tool to quantify plaque vulnerability [32]. Indeed, plaque echolucency is the imaging correlated to histopathologic evidence of either lipid-rich necrotic core and/or intraplaque hemorrhage [33,34]. Echolucent plaques have been shown to have an increased risk of ischemic cerebrovascular events. In the Tromsø study, patients with echolucent atherosclerotic plaques had an increased risk of ischemic cerebrovascular events, independent of the degree of stenosis and cardiovascular risk factors [35]. The international multicenter Imaging in Carotid Angioplasties and Risk of Stroke registry also showed an increase in stroke risk in carotid artery stenting in patients with a GSM < 25 [14]. Our study provides the first evidence that carotid echolucent plaques, as measured by GSM, are associated with worse cognitive performance and a more rapid progression of cognitive dysfunction in comparison to subjects with iso- or hyperechogenic ones.

From a pathophysiological perspective, it is conceivable that the increased risk of ischemic cerebrovascular events associated with plaque echolucency could have contributed to our findings. Anyway, none of the patients included in the analysis had a history or clinical evidence of cerebrovascular disease, either at baseline or during the follow-up. In addition, among patients included in the baseline analyses who underwent carotid endarterectomy because of severe stenosis, none had neuroradiological evidence of vascular brain damage. Obviously, we cannot exclude the eventual occurrence of silent cerebrovascular damage in some subjects with echolucent plaques. In this regard, there is evidence that vulnerable plaque features contribute to cortical micro-infarcts detected on MR, which are, in turn, associated with poor cognitive functions [36]. In addition, it is possible that both plaque echolucency and cognitive impairment could be expressions of a common inflammatory pathway potentially responsible for the progression of both cognitive impairment and vascular damage [37,38,39]. Notably, the neuropsychological tests we used in the current report have been designed to explore mainly executive functions, which are known for being deeply influenced by vascular factors [40]. On the other hand, our data seem to exclude any relevant role for hemodynamic factors since the relationship between the echogenic feature of carotid plaque and cognitive performance was not influenced by the degree of carotid stenosis as measured by NASCET.

The potential clinical relevance of our results requires some consideration. First, we have to underline that the follow-up analysis was performed in about half of the initial cohort. Thus, we cannot exclude a potential bias deriving from the low number of subjects re-analyzed at follow-up. This consistent reduction of patients during follow-up does not allow speculations about the potential determinants of the cognitive dysfunction other than the assessment of the cross-sectional independent association between the variables evaluated at the two-time points. In this regard, the inverse association between plaque echolucency and cognitive functions observed at baseline was still evident at follow-up. In addition, among patients re-evaluated during follow-up, the composite cognitive z-score significantly decreased in those with echolucent plaques but not in those with iso- or hyperechogenic plaques. This finding further reinforces the hypothesis of an influential role of echogenic features of the plaques on cognitive function. We also have to consider some potential limitations of the GSM approach used to assess plaque echogenicity. One is that GSM is a median value of pixel brightness of the entire carotid plaque, not taking into consideration focal variability [41]. The GSM of a homogeneous gray field and the GSM of a checkboard will theoretically be equal, although the pattern is different [41]. Another limitation in the use of the GSM is the subjectivity in outlining the plaques before GSM analysis is performed, although studies have shown reasonable intra- and interobserver variability [41]. Despite these potential limitations, it should be considered that ultrasound is a readily available, relatively inexpensive, and widely used diagnostic method.

## 5. Conclusions

The present study demonstrated that carotid plaque echolucency is associated with worse cognitive performance and more rapid progression of cognitive dysfunction in elderly subjects with atherosclerotic disease. Although these data require confirmation from larger studies, they suggest that the assessment of plaque echogenicity, if used appropriately, might aid in identifying subjects at increased risk for cognitive dysfunction.

## Figures and Tables

**Figure 1 metabolites-13-00478-f001:**
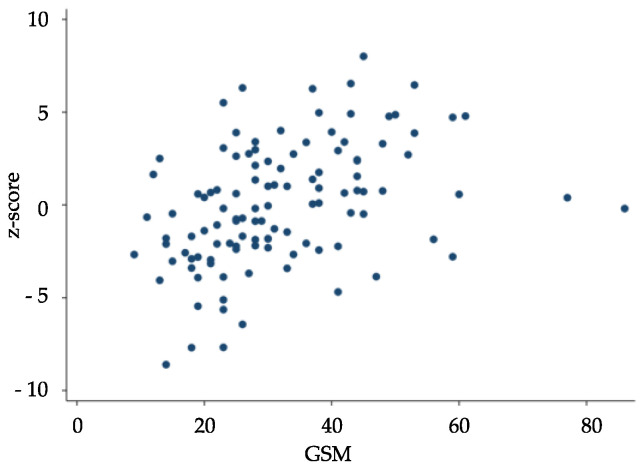
Spearman’s correlation between gray-scale median (GSM) and z-score at baseline (rho = 0.464; *p* < 0.0001).

**Figure 2 metabolites-13-00478-f002:**
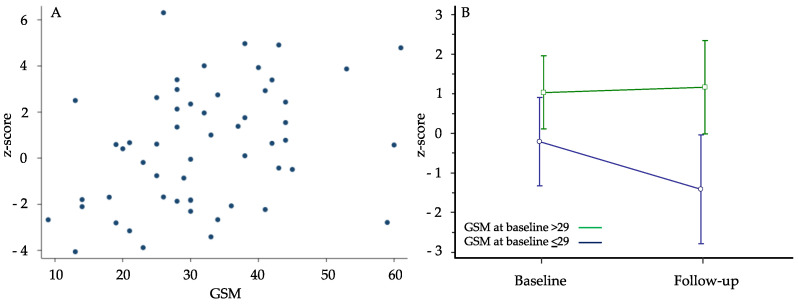
Panel (**A**) Spearman’s correlation between gray-scale median (GSM) and z-score at follow-up (rho: 0.392; *p* = 0.003). Panel (**B**) variation of composite z-score in 2 groups of participants distinguished on the basis of the median GSM value of 29 assessed at first examination (two-factor analysis of variance—ANOVA with repeated measures over time).

**Table 1 metabolites-13-00478-t001:** General characteristics of the study population at baseline.

Age	72.4 ± 5.9
Gender (males/females)	59/54
BMI (kg/m^2^)	27.5 ± 4.1
Total cholesterol (mg/dL)	198.2 ± 47.0
HDL-C (mg/dL)	48.1 ± 12.4
LDL-C (mg/dL)	119.5 ± 44.2
Triglyceride (mg/dL)	154.9 ± 68.1
SBP (mmHg)	138.8 ± 15.2
DBP (mmHg)	78.3 ± 8
GSM	32.1 ± 13.7
NASCET (%)	43.6 ± 20.8
Hypertension (%)	91 (80.5%)
Hypercholesterolemia (%)	83 (73.5%)
Diabetes mellitus (%)	22 (19.5%)
Smoking (%)	17 (15.0%)
MMSE score	27.5 ± 1.8
TMT-A (seconds)	78.6 ± 35.6
TMT-B (seconds)	178.6 ± 78.3
TMT B-A (seconds)	99.9 ± 57.2
VFT (n. words)	21.4 ± 7.8
z-score	0.01 ± 3.3

BMI: body mass index; HDL: high-density lipoprotein cholesterol; LDL: low-density lipoprotein cholesterol; SBP: systolic blood pressure; DBP: diastolic blood pressure; GSM: gray-scale median; NASCET: North American Symptomatic Carotid Endarterectomy Trial; MMSE: Mini Mental State Examination; TMT: Trail Making Test; VFT: Verbal Fluency Test.

**Table 2 metabolites-13-00478-t002:** Linear regression analysis for prediction of z-score at baseline.

	Coefficient	95% C.I.	*p*
Age	−0.1514065	−0.25; −0.05	0.003
Hypertension	−0.1155479	−1.57; 1.34	0.875
Hypercholesterolemia	0.5500091	−0.73; 1.83	0.397
Diabetes mellitus	0.2200304	−1.17; 1.61	0.755
BMI	−0.1115516	−0.25; 0.03	0.113
GSM	0.0954297	0.05; 0.14	<0.0001
NASCET	0.0003074	−0.03; 0.03	0.983

BMI: body mass index; GSM: grayscale median; NASCET: North American Symptomatic Carotid Endarterectomy Trial.

**Table 3 metabolites-13-00478-t003:** Neuropsychological test scores in relation to the median GSM value at baseline.

	GSM ≤ 29 n = 57	GSM > 29 n = 56	*p* *
MMSE score	27.2 ± 1.8	27.8 ± 1.7	0.078
TMT-A (seconds)	91.2 ± 34.4	65.8 ± 32.2	<0.0001
TMT-B (seconds)	205.1 ± 84.9	151.6 ± 60.7	<0.0001
TMT B-A (seconds)	113.8 ± 67.9	85.8 ± 39.5	0.0178
VFT (n. words)	18.9 ± 7.4	24.0 ± 7.4	0.0002
z-score	−1.3 ± 3.2	1.3 ± 2.9	<0.0001

* Wilcoxon rank-sum test. MMSE: Mini Mental State Examination; TMT: Trail Making Test; VFT: verbal fluency test.

**Table 4 metabolites-13-00478-t004:** Neuropsychological test scores at baseline and follow-up.

	Baseline n = 55	Follow-Up n = 55	*p* *
MMSE score	27.8 ± 1.7	27.7 ± 2.5	0.8710
TMT-A (seconds)	73.9 ± 30.8	75.0 ± 33.7	0.9365
TMT-B (seconds)	162.2 ± 52.8	193.4 ± 110.1	0.0103
TMT B-A (seconds)	88.3 ± 39.9	118.3 ± 83.8	0.0065
VFT (n. words)	21.7 ± 7.3	20.3 ± 7.2	0.0899
z-score	0.5 ± 2.6	0.1 ± 3.44	0.2580

* Wilcoxon signed-rank test. MMSE: Mini Mental State Examination; TMT: Trail Making Test; VFT: verbal fluency test.

**Table 5 metabolites-13-00478-t005:** Linear regression analysis for prediction of z-score at follow-up.

	Coefficient	95% C.I.	*p*
Age	−0.21112	−0.37; −0.05	0.013
Hypertension	−2.39924	−5.19; 0.39	0.09
Hypercholesterolemia	0.753556	−1.42; 2.93	0.489
Diabetes mellitus	0.694924	−1.19; 2.58	0.462
BMI	0.054576	−0.16; 0.27	0.618
GSM	0.1117	0.03; 0.19	0.006
NASCET	0.039326	−0.04; 0.12	0.316

BMI: body mass index; GSM: gray-scale median; NASCET: North American Symptomatic Carotid Endarterectomy Trial.

## Data Availability

Data are available upon reasonable request to the investigators. The data are not publicly available due to specific restrictions.

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
