# Peer review of "Enhanced Carotid Plaque Echolucency Is Associated with Reduced Cognitive Performance in Elderly Patients with Atherosclerotic Disease Independently on Metabolic Profile"

_metabolites, 2023, doi:10.3390/metabo13040478_

Round 1
Reviewer 1 Report
Dear Authors, your study presents some important limitations needing your explanation:
- Numbers of patients is very low, unsufficient to outline significant outcomes;
- Did you perform GSM score in all pts without problems? In some cases with important echolucency have you been difficulties?
- Did you correlated wit intracranial circulation? Have you performed MRI or Angio-CT scan in order to analyze brain circulation and presence of calcification or silent ischemic lesions at that level?
-50% of lost of follow-up is very high, Why?
Author Response
Re: Metabolites-2245529 - “Enhanced carotid plaque echolucency is associated with reduced cognitive performance in elderly patients with atherosclerotic disease independently on metabolic profile”.
Reviewer #1
Dear Sir/Madam
Thank you very much for your appreciation of our work. As you can see the manuscript has been revised according to your kind comments and fruitfull suggestions.
Numbers of patients is very low, unsufficient to outline significant outcomes
Reply: thank you for this observation. Due to the lack of previous evidence on the topic of our study, we estimated a sample size allowing us to identify any clinically significant correlation in the cross-sectional phase of the study. The evidence of significant correlations between plaque echolucency and cognitive scores support our estimation. Anyway, the relatively low number of patients somewhat limits the generalizability of our results. Please note that have added this sentence as a potential study limitation and we have softened our conclusions.
Did you perform GSM score in all pts without problems? In some cases with important echolucency have you been difficulties?
Reply: we appreciate this observation. In order to guarantee the correct delineation of the plaque borders, we saved one image with colour or power Doppler in patients with hypoechoic or anechoic plaques. Successively, we omitted the colour information and calculated GSM using grey mode images. Please note that we have further clarified this aspect in the revised version of our manuscript.
Did you correlated wit intracranial circulation? Have you performed MRI or Angio-CT scan in order to analyze brain circulation and presence of calcification or silent ischemic lesions at that level?
Reply: we did not systematically correlate our findings with transcranial doppler ultrasonography nor with MRI or Angio-CT scans. However, all patients with plaques causing severe stenosis (70% or more according to NASCET method) underwent CT brain scans prior to endoarterectomy, in order to exclude recent ischemic lesions. In this subgroup of patients, all subjects with evidence of ischemic brain lesions were excluded from the analyses. Please note that we have better explained this aspect in methods section. In addition, we discussed this relevant topic in discussion section. Thank you also for this suggestion.
50% of lost of follow-up is very high, Why?
Reply: as it is now better explained in the method section, 32 subjects after the first evaluation underwent carotid endarterectomy because of severe carotid stenosis (70% or more according to NASCET method). These patients were excludes from follow-up. Among the remaining patients kept on medical treatment, 12 declined to attend the follow-up evaluation and 6 died while 8 were excluded from follow-up because they experienced an acute vascular event. Please note that we have better explained this aspect in the results section. In addition, it has been better discussed as potential study limitation. Thank you also for this suggestion.
We hope that this revised version of our work could encounter your appreciation.
I look forward to hearing from you.
Yours sincerely,
Giovambattista Desideri

Reviewer 2 Report
The authors describe an interesting study regarding the association between echogenicity of carotid plaques and cognitive performance in patients with carotid atherosclerotic plaques and no history of cerebrovascular events and/or clinical evidence for dementia. Overall, the article is well-written.
However, there are some points, that should be addressed:
-) The authors said that the degree of stenosis was measured according to the NASCET method. Will be very interesting if the authors cand provide the degree of stenosis at follow up and identify the risk factors that are associated with the increase of stenosis degree.
-) The authors can improve the Discussion section by citing the following articles regarding the atherosclerotic plaque and poor outcome:
-https://doi.org/10.3390/ijerph192113934
-https://doi.org/10.3390/diagnostics13010142
-) There are some irregularities and spelling errors across the manuscript, which should be further edited
Author Response
Re: Metabolites-2245529 - “Enhanced carotid plaque echolucency is associated with reduced cognitive performance in elderly patients with atherosclerotic disease independently on metabolic profile”.
Reviewer #2
Dear Sir/Madam,
Thank you very much for your appreciation of our work. As you can see the manuscript has been revised according to your kind comments and fruitfull suggestions.
The authors said that the degree of stenosis was measured according to the NASCET method. Will be very interesting if the authors cand provide the degree of stenosis at follow up and identify the risk factors that are associated with the increase of stenosis degree.
Reply: we appreciate this observation. A significant variation of NASCET value was observed in the subgroup of patient followed during the longitudinal phase of the study (from 32.1+12.1 to 36.9+13.1, p<0.0001). Significant correlations between NASCET values and both systolic and diastolic blood pressure were found at baseline (appendix A) but the independency of these associations was not confirmed by multiple linear regression analysis (systolic blood pressure, b=0.221, p=0.137; diastolic blood pressure b=0.264, p=0.358). These correlations were no more evident at follow-up likely because the low number of subject re-analyzed at this time point. The consistent reduction of patients during follow-up does not allow speculations about the potential determinants of the NASCET values other than the assessment of the cross-sectional independent association at the 2 time points. Please note that we have added data on variation of NASCET in results section and modified the discussion accordingly.
The authors can improve the Discussion section by citing the following articles regarding the atherosclerotic plaque and poor outcome:
-https://doi.org/10.3390/ijerph192113934
-https://doi.org/10.3390/diagnostics13010142
Replies: Thank you for this suggestion. Both these quoted articles have been cited and briefly discussed.
There are some irregularities and spelling errors across the manuscript, which should be further edited
Replies: Thank you for this suggestion. The manuscript has been revised by a mother tongue colleague.
We hope that this revised version of our work could encounter your appreciation.
I look forward to hearing from you.
Yours sincerely,
Giovambattista Desideri

Round 2
Reviewer 1 Report
Dear Authors,
I have not more concerns
Reviewer 2 Report
no further comments